# Experimental Study on Electrochemical Desulfurization of Coal Liquefaction Residue

**DOI:** 10.3390/molecules28062749

**Published:** 2023-03-18

**Authors:** Jianming Fan, Yongfeng Zhang, Na Li, Ruzhan Bai, Qi Liu, Xing Zhou

**Affiliations:** 1College of Chemical Engineering, Inner Mongolia University of Technology, Huhhot 010051, China; 2Department of Chemical Engineering, Ordos Vocational College, Ordos 017010, China; 3Inner Mongolia Key Laboratory of Efficient Recycle Utilization for Coal-Based Waste, Huhhot 010051, China; 4Inner Mongolia Key Laboratory of High-Value Functional Utilization of Low Rank Carbon Resources, Huhhot 010051, China; 5College of Chemistry and Chemical Engineering, Taiyuan University of Technology, Taiyuan 030024, China; 6Hebei Key Laboratory of Inorganic Nanomaterials, School of Chemistry and Materials Science, Hebei Normal University, Shijiazhuang 050024, China; 7College of Zhongran, Hebei Normal University, Shijiazhuang 050024, China

**Keywords:** coal liquefaction residue, electrolytic chemistry, desulfurization, energy consumption, structural changes

## Abstract

The occurrence of sulfur in coal direct liquefaction residue affects its further high quality and high value utilization. Electrochemical desulfurization is characterized by mild reaction conditions, simple operation, easy separation of sulfur conversion products and little influence on the properties of the liquefied residue. An anodic electrolytic oxidation desulphurization experiment was carried out on the liquefaction residue of the by-product of a coal-to-liquid enterprise in the slurry state. An electrochemical test and material characterization of raw materials before and after electrolysis showed that electrolytic oxidation can desulfurize the liquefaction residue under an alkaline condition. Linear sweep voltammetry (LSV) was used for the electrolysis experiments to obtain the optimal slurry concentration of 60 g/L. On this basis, the reaction kinetics were calculated, and the minimum activation energy in the interval at 0.9 (V vs. Hg/HgO) was 19.71 kJ/mol. The relationship between the electrolytic desulfurization of the liquefied residue and energy consumption was studied by the potentiostatic method. The influence of anodic potential and electrolytic temperature on the current density, cell voltage, desulfurization rate and energy consumption was investigated. The experimental results showed that the desulfurization rate and total energy consumption increase positively with the increase in reaction temperature and electrolytic potential in a certain range. The influence of the reaction temperature on the desulfurization rate and total energy consumption is more prominent than that of electrolytic potential, but the energy consumption of sulfur removal per unit mass does not show a positive correlation. Therefore, with the energy consumption per unit mass of sulfur removal as the efficiency index, the optimal experimental results were obtained: under the conditions of 0.8 (V vs. Hg/HgO) anode potential, 50 °C electrolytic temperature, 60 g/L slurry concentration and 14,400 s electrolytic time, the desulfurization rate was 18.85%, and the power consumption per unit mass of sulfur removal was 5585.74 W·s/g. The results of XPS, SEM, BET and IC showed that both inorganic and organic sulfur were removed by electrolytic oxidation, and the morphology, pore structure and chemical bond of the liquefied residue were affected by electrolytic oxidation. The research method provides a new idea and reference for the efficiency evaluation of desulfurization and hydrogen production from coal liquefaction residue.

## 1. Introduction

Direct coal liquefaction is the technology of catalyzing the conversion of coal to clean liquid coal through hydrocracking of the polycyclic aromatic hydrocarbons in coal to small aliphatic molecules [1], which is an important component and typical demonstration of modern coal chemical technology. Coal liquefaction residue, also known as coal liquefaction asphalt, is a bulk by-product produced by decompression distillation of hydrogenation liquefaction products in the process of direct coal to oil production. It has the characteristics of high carbon residue, high ash and high sulfur [2,3]. It has been separated by vacuum distillation, accounting for about 20~30 wt.% of the total liquefied raw coal [4,5]. At present, the production enterprises sell and dispose of it as a general solid waste and the resource utilization rate and economy are low.

The main components of coal liquefaction residue include: unconverted raw coal, inorganic minerals, residual catalysts, heavy oil, asphaltene, etc. [6,7]. At present, the research on liquefaction residue mainly focuses on the gasification [5,8,9,10], pyrolysis [11,12,13], combustion [14] and preparation of functional carbon materials [15,16,17]. However, it is not hard to find a common feature in applied research in these fields. Pyrolysis is a fundamental and indispensable stage of most thermal conversion processes. The occurrence of high sulfur in coal liquefaction residue inevitably brings adverse effects [18]. Sulfur will spilt in the form of a gas phase and the next step of the gas phase sulfur removal, and separation will increase its application cost. It is because of this adverse factor that desulfurization is an essential process for high-quality and high-value utilization of liquefied residue.

The occurrence of sulfur in coal liquefaction residue mainly includes two types: inorganic sulfur and organic sulfur. The main form of inorganic sulfur is pyrite. This is related to the sulfide reaction between the iron precursor and the injected sulfur to form the pyrite phase during coal liquefaction. Organic sulfur mainly contains thiol, thiamidine and thiophene. These organic sulfur materials are mainly derived from raw coal. The proportion of inorganic sulfur is generally small; it is mainly organic sulfur. The type of coal liquefaction residue is similar to the type of sulfur in coal, so the coal liquefaction residue can also be studied according to the research method of sulfur in coal. Currently, the desulfurization technologies suitable for coal can generally be divided into physical desulfurization [19], chemical desulfurization [20] and biological desulfurization [21]. Traditional physical desulfurization methods, such as flotation technology, can only remove part of the inorganic sulfur in coal, and then, organic sulfur cannot be removed. The condition of biological desulphurization is harsh, and the chemical method is more suitable for deep desulphurization. The coal liquefaction residue is produced in dense flake form, which limits the application of physical desulfurization technology, such as flotation technology [22]. Organic sulfur is suitable for chemical removal by thermochemical oxidation to convert organic sulfur to sulfur-containing gases such as SO_2_ and SO_3_, or by electrochemical reduction to H_2_S and S^2−^ [23,24], and by electrochemical oxidation to convert organic sulfur to soluble sulfonic acid and sulfate [25].

However, the process route of the thermochemical oxidation desulfurization method is long, the operating conditions are complex, the separation of sulfur-containing gas will increase the economic burden and the high temperature conditions change the original characteristics of the liquefaction residue, which will limit its subsequent application range. There are fewer reducing agents for electrochemical reduction desulfurization, for example, H_2_ and NaBH_4_ have been studied [26,27]. At the same time, the reduction reaction process will produce H_2_S, which is a kind of harmful gas, resulting in environmental pollution [28,29] and the desulfurization cost is relatively high. In contrast, electrochemical oxidation desulfurization has mild reaction conditions, simple operation [28,30], easy separation of sulfur conversion products [31] and little influence on the characteristics of the liquefaction residue.

Electrochemical oxidation desulphurization is usually carried out in slurry [32,33]. The effect is more prominent with the use of an alkaline electrolyte [34,35] and the electrolyte is less harmful to the electrode material. Electrochemical desulfurization under an alkaline environment with both alkali leaching [32] and electrochemical oxidation [36] can enhance the removal effect of inorganic sulfur and organic sulfur in coal liquefaction residue. Alkaline system electrolytic oxidation desulfurization mainly occurs on the anode. The oxidation group ROSs (such as HO•, O•, O_2_^−^•, O_2_, etc.) produced on the anode are used as a strong oxidant. Inorganic sulfur is oxidized to soluble sulfate [37,38], organic sulfur is oxidized to sulfoxide, sulfoxide is further oxidized to sulfone and sulfone can be hydrolyzed to soluble sulfonic acid and sulfate. Soluble sulfate and sulfonic acid can be removed by filtration. It is easy to see that sulfur in the coal liquefaction residue in the alkaline system is mainly oxidized to SO_4_^2−^ to achieve desulfurization, and high purity H_2_ is produced at the cathode as a by-product [31]. This method has high comprehensive application.

In the basic system, the following reactions mainly occur in the electrochemical desulfurization process [39]:

Principle of inorganic sulfur (FeS_2_) desulfurization by anodic oxidation:2 H_2_O → O_2_ + 4 H^+^ + 4 e^−^(1)
16 OH^−^ + 4 FeS_2_ + 15 O_2_ → 4 Fe(OH)_3_ + 8 SO_4_^2−^ + 2 H_2_O (2)
8 OH^−^ + 2 FeS_2_ + 7 O_2_ → 2 Fe(OH)_2_ + 4 SO_4_^2−^ + 2 H_2_O (3)
FeS_2_ + 6 HO· → Fe(OH)_3_ + S_2_O_3_^2−^ + 3 H_2_O + 2 e^−^(4)
2 FeS_2_ + 3 O_2_^−^ → 2 Fe^3+^ + 2 S_2_O_3_^2−^ + 5 e^−^
(5)

Principle of organic sulfur desulfurization by anodic oxidation:O_2_ + 2 R-S-S-R → 2 R-S-S(O)-R (6)
2 O_2_ + R-S-S-R → R-S(O_2_)-S(O_2_)-R (7)
R-S(O_2_)-S(O_2_)-R + 2 H_2_O → R-OH + R-OH + 2 SO_4_^2−^
(8)

With the wide application of new energy power generation technologies, cheap renewable electricity can be used as desulfurization energy; thus, the cost of the electricity used by electrochemical methods can be effectively reduced. However, in the process of practical application or industrialization, the energy consumption of electrochemical methods remains high. Therefore, the basic data related to energy consumption are of great significance for the large-scale application of electrochemical oxidation and desulfurization. However, the relationship between desulfurization energy consumption and desulfurization efficiency is often ignored in many studies. For example, Zhang et al. investigated the influence of pressure enhancement factors and the design of high temperature electrolytic process conditions on the rate of electrolytic desulfurization [40,41]. Han et al. paid attention to the influence of electrolytic temperature and current value on the desulfurization rate and efficiency [42]. Gong et al. investigated the influence of different electrolytes and slurry stirring and reaction temperature on the desulphurization rate [43]. However, there is a lack of attention on the important index of desulfurization energy consumption per unit mass. Similar studies have focused on the influence of experimental methods on desulfurization efficiency.

Desulfurization energy consumption, desulfurization efficiency and sample structure changes in the electrolytic desulfurization process of coal liquefaction residue are very important for the popularization and application of electrochemical oxidation and desulfurization methods. Therefore, this paper uses electrochemical means such as LSV and constant potential electrolysis to study the electrolytic desulfurization characteristics of coal liquefied residue. The results show that the electrolytic oxidation method can remove both inorganic sulfur and organic sulfur from coal liquefaction residue. The desulfurization energy consumption and desulfurization efficiency are associated with the reaction conditions. The energy consumption per unit mass of sulfur removal is innovatively proposed as the efficiency index, and the experimental data of the power consumption per unit mass of sulfur removal is obtained. Combined with SEM, BET, XPS and other means, the sample structures before and after electrolysis are characterized. It provides a reference for the efficiency evaluation of desulfurization and hydrogen production from coal liquefaction residue.

## 2. Results and Discussion

### 2.1. LSV Polarization Curve

Figure 1a shows the LSV polarization curve of different slurry concentration conditions of the coal liquefaction residue. As can be seen from the (a) diagram, with the increase in slurry concentration, a higher current density is obtained after adding coal liquefaction residue in the 0.7–0.9 (V vs. Hg/HgO) range, indicating that the liquefaction residue in the slurry participates in the anode reaction. This potential interval is consistent with the oxidation potential of the S atom. During the experiment, the serous concentrations of 40 g/L and 60 g/L in the process achieved good results. Figure 1b shows the LSV polarization curve of 60 g/L at different electrolytic temperatures. It can be seen that the temperature has a more obvious effect on the electrolysis reaction of the coal liquefaction residue. In the 0.7–0.9 (V vs. Hg/HgO) interval, the current density value increases correspondingly with the increase in the electrolytic temperature under the same potential.

### 2.2. Reaction Dynamics

The current density reflects the electrode reaction rate, and the electrode reaction rate is closely related to the size of its activation energy. The lower the activation energy, the faster the reaction rate. Therefore, calculating the reaction activation energy is very important for accumulating the basic data of the electrolytic desulfurization of liquefaction residue.

Convert Formula (11) to logarithmic form to obtain Formula (9).
(9)$$\mathrm{lg}j=\mathrm{lg}f-\frac{E_{a}}{2.303RT}$$

Formula (9) indicates that there is a linear relationship between the sum and the slope is $-\frac{E_{a}}{2.303RT}$. According to Figure 1b, with the LSV curve at different temperatures (30–70 °C), the relationship between lgj and 1/T is obtained. On this basis, linear fitting was carried out, as shown in Figure 2a. Then, according to its slope, the dynamic curve was drawn, as shown in Figure 2b.

From Figure 2b, it can be seen that the apparent activation energy of the electrolysis of coal liquefaction residue slurry decreases with the increase in potential value. There is an interval minimum activation energy $E_{a}$ = 19.71 KJ/mol at 0.9 (V vs. Hg/HgO). The result is consistent with the maximum current density obtained at 0.9 (V vs. Hg/HgO) in the potentiostatic electrolysis experiment.

### 2.3. Electrolytic Desulfurization Efficiency and Desulfurization Energy Consumption

Taking an electrolytic temperature of 30 °C and slurry concentration of 60 g/L as the research objects, Figure 3 examines the relationship among the current density, cell voltage, desulfurization rate and energy consumption in the constant potential electrolytic desulfurization experiment under different anode potentials. As can be seen from Figure 3a, both the desulfurization rate and total energy consumption show an upward trend with the increase in electrolytic potential, and the change in total energy consumption is larger than that of the desulfurization rate. The increase rate of energy consumption caused by the increase of 0.8 (V vs. Hg/HgO) to 0.9 (V vs. Hg/HgO) is 2.93 times that of 0.7 (V vs. Hg/HgO) to 0.8 (V vs. Hg/HgO), but the increase rate of desulfurization decreases by 3.7% under the same conditions. Therefore, taking the energy consumption per unit mass of sulfur removal as the efficiency index, the minimum energy consumption of 6950.63 W·s/g was obtained at the potential value of 0.8 (V vs. Hg/HgO).

It can be seen from Figure 3b that the increase in the potential value also leads to the positive increase in the current density and the cell voltage, but the change in the current density value is more obvious than that of the cell voltage. With the increase in electrolytic potential, the oxygen evolution reaction at the anode becomes more prominent, and the electrode reaction becomes more intense, resulting in a higher current density value. The increase in electrolytic potential also leads to a corresponding increase in the amount of ROSs produced at the anode. However, for the liquefaction residue mainly in the form of organic sulfur, the contact between organic sulfur in the particle and ROSs is limited, and the increasing potential still has a weak effect on the sulfur removal effect of the sample. It can be seen from Figure 3a that with the increase in electrolytic potential, the desulfurization rate is still low, and the maximum desulfurization rate is 15.84%.

On the basis of the electrolytic potential experiment, taking 60 g/L slurry concentration as the research object, the influence of temperature on the electrolytic efficiency and energy consumption was further investigated under the electrolytic potential of 0.8 (V vs. Hg/HgO). Considering the influence of temperature on the Hg/HgO reference electrode, the electrolytic reaction experiment at a higher temperature was not conducted, and the results are shown in Figure 4. It can be seen from Figure 4 that the temperature has a more obvious influence on the electrolytic desulfurization reaction. With the increase in the electrolytic reaction temperature, the desulfurization rate and the total energy consumption show an increasing trend. With the increase in reaction temperature, the reaction rate of the electrode increases continuously, which is manifested by the large increase in current density value, with an increase of 10.10% at 50 °C compared with 30 °C and 41.72% at 70 °C compared with 50 °C. However, the temperature has little effect on the cell voltage. It was found that there was no positive correlation between the energy consumption per unit mass of sulfur removal and the reaction temperature. Therefore, taking the energy consumption per unit mass of sulfur removal as the efficiency index, the minimum energy consumption of 5585.74 W·s/g was obtained at the electrolytic temperature of 50 °C.

For slurry reaction, on the one hand, the irregular movement of liquefaction residue particles is intensified with the increase in temperature, which leads to the increase in the contact rate with the anode electrode surface and increases the direct oxidation opportunity of the sulfur phase. On the other hand, the increase in temperature is beneficial to increase the formation rate of ROSs, which promotes the transfer of ROSs and increases the chance of collision between ROSs and sulfur, which is beneficial to the indirect oxidation reaction between ROSs and sulfur. Increasing the reaction temperature directly affects the current density value, which indirectly leads to the continuous increase in the total energy consumption. Therefore, in order to obtain a larger current density value, it is necessary to increase the energy consumption at the cost, and the effective utilization of energy consumption is the basis for judging the optimal reaction temperature.

### 2.4. Mechanism Analysis of Electrolytic Oxidation Desulfurization

#### 2.4.1. Chemical Bond Analysis of Liquefied Residue

Figure 5 shows the XPS S *2p* and C *1s* spectra before and after electrolysis of the coal liquefaction residue. Table 1 shows the relative content analysis of S *2p* and C *1s* before and after electrolysis of the coal liquefaction residue. It can be seen that the structure of S and C of the coal liquefaction residue changes before and after electrolysis. Before electrolysis, the coal liquefaction residue mainly exists in the chemical state of satellites (pyrite), thiophene (thiophene), R-S-R (thiamidine) and R-SH (thiol). After electrolysis, the contents of satellites (pyrite), thiophene (thiophene), R-S-R (thiamidine) and R-SH (thiol) all decrease, indicating that both inorganic and organic sulfur in the coal liquefaction residue are electrolytic oxidized.

However, from the relative content change values in Table 1, it can be seen that inorganic sulfur and organic sulfur in the coal liquefaction residue are only partially electrolytic oxidized, which is consistent with the lower desulfurization rate measured in the experiment. Therefore, for the liquefied residue hydrophobic sulfur-containing substances, the existence of adhesion agglomeration between solid particles results in poor dispersion uniformity, and the dense structure of sulfur in the deep is difficult to removes; as a result, the electrochemical slurry desulphurization must have the characteristic of a low desulphurization rate. Therefore, the slurry desulphurization method is more suitable for desulphurization pretreatment under the condition of effective energy consumption utilization. It was also found that the contents of the C-C bond and the O-C = O bond decreased by 0.88% and 3.04%, respectively, while the content of the C-O bond increased by 12.15%, showing that the oxidation reaction of part of the C structure also occurs in the process of electrolytic desulphurization, and this reaction increases part of the energy consumption.

#### 2.4.2. Surface Morphology and Pore Structure of Liquefied Residue

The sulfur in the liquefaction residue exists in the form of inorganic pyrite and organic sulfur such as thiophene, thiamidine, etc. Figure 6 shows the changes in the surface morphology before and after electrolysis, respectively. Figure 7 shows the N_2_ adsorption–desorption curves and pore distribution of the samples, and the corresponding specific surface area and pore size data are shown in Table 4. As can be seen from Figure 6, the morphology of the particles before and after electrolysis is irregular, mainly in the form of a lamellar bulk structure, which is relatively compact, and the lamellar structure has denudation after electrolysis, which may be caused by the oxidation of sulfur and carbon on the surface. The results show that the electrolytic process causes some damage to the surface morphology of the liquefaction residue particles but has little effect on its original structure.

From Figure 7, the adsorption types of the samples are all type II; all of them have a H3 hysteresis loop (according to IUPAC classification). Capillary condensation occurs at a slightly higher pressure, which indicates that there are some larger mesoporous pores in the samples. From Table 2, the specific surface area and pore volume of the liquefaction residue particles after electrolysis increased by 69.81% (2.78 m^2^/g) and 46.13% (0.00234 m^3^/g), respectively, but the pore size decreased by 27.33% (2.2152 nm). The results show that the electrolysis process has obvious influence on the pore structure of the liquefaction residue particles. The main causes of the above changes are analyzed. The surface and interior pyrite and organic sulfur of the liquefaction residue are partly oxidized by the electrolytic process, and the carbon of the liquefaction residue is also partly consumed by oxidation. The oxidation loss of these substances leads to the addition of more fine pores, which increases the specific area and pore volume of particles and decreases the average pore size.

#### 2.4.3. Electrolyte Ion Analysis

Based on the principle of the desulphurization reaction under alkaline conditions, the SO_4_^2−^ directional detection of electrolyte filtrate can help to judge the desulfurization oxidation, and the results are shown in Figure 8. The SO_4_^2−^ peak appears between 20 and 25 min and the concentration of SO_4_^2−^ is 37.94 mg/L after diluting the electrolyte. The results show that SO_4_^2−^ is formed in the process of electrolytic oxidation, which indicates that oxidation of sulfur occurs.

## 3. Experimental Part

### 3.1. Raw Material

The residue (SH-DCLR), which is flaky, from the direct coal liquefaction of a coal chemical company was selected as the experimental raw material, and the material was used in the experiment after crushing, grinding and screening. Table 3 shows proximate analysis and ultimate analysis of the raw material samples. Table 4 shows measurement results of the sulfur distribution in raw material samples. It can be seen from sample test results that raw material of SH-DCLR liquefaction residue has a high carbon content and great utilization value; the sulfur content reaches the medium sulfur content, which is mainly composed of inorganic sulfur (FeS_2_) and organic sulfur, and the content of inorganic sulfur (FeS_2_) is relatively low.

### 3.2. Experimental Apparatus and Methods

The three electrode system was used for electrochemical testing. Both the working electrode and counter electrode are platinum (Pt:10 × 10 × 0.1 mm, effective area 2 cm^2^). Hg/HgO is used as reference electrode, and the electrolyte is 1 M KOH. The electrolyzer is heated by circulating water.

#### 3.2.1. Desulfurization Rate and Kinetic Calculation

Sulfur content of samples was determined by Coulomb legal sulfur meter for analysis and the formula for calculating the desulfurization rate is shown in (10)
(10)$$\rho =\frac{S_{0}-S_{1}}{S_{0}}\times 100$$

In the formula:

ρ—Desulfurization rate, %;S_0_—S content in raw material liquefaction residue, %;S_1_—S content in liquefaction residue after electrolysis, %.

Kinetic calculation According to Arrhenius Equation, the relationship between reaction rate and activation energy of electrode reaction is expressed as Equation (11) [24].
(11)$$j=f\exp(-\frac{E_{a}}{RT})$$

In the formula:

j—current density, A/cm^2^;E_a_—activation energy, J/mol;f—pre-exponential factor;R—gas molar constant, 8.314 J/mol·K;T—thermodynamic temperature, K.

#### 3.2.2. Calculation of Energy Consumption of Per Unit Sulfur Removal


(12)
$$E=\frac{\sum U^{\prime }\times I\times t}{m_{s}}$$


In the formula:

E—cumulative electricity consumption, W·s/g;U’—real-time electrolyzer voltage, V;I—real-time current, A;t—electrolysis time, 1 s;m_s_—sulfur removal amount, g.

### 3.3. Experimental Characterization Methods

The ESCLAB-250 Xi type X-ray photoelectron spectrometer (XPS) produced by Thermo-Fisher Company of the United States was used to analyze the sulfur state of the surface elements of the samples. The monochromatic X-ray source Al anode target (1486.6 eV), the beam spot diameter: 200 μm. The Apreo C type scanning electron microscope (SEM) by Thermo-Fisher Company of the United States was used to analyze the surface morphology of the samples. Test conditions: accelerated voltage of 10 kV, backscattered electron imaging. Nitrogen adsorption and desorption of samples were tested by ASAP2460 multi-station extended specific surface and porosity analyzer from Micromeritics of the United States, and the specific surface area, pore volume and pore diameter of samples were obtained. Cic-100 ion chromatograph (IC) was used for SO_4_^2−^ directional detection of electrolyte filtrate.

## 4. Conclusions

The single factor experimental design method was used to carry out the experimental research of constant potential desulfurization and electrolysis. The results show that the electrolysis temperature, anode potential and current density have great effects on the desulfurization efficiency and energy consumption. The desulfurization rate and total energy consumption increase positively with the increase in reaction temperature and electrolytic potential in a certain range. The influence of the reaction temperature on the desulfurization rate and total energy consumption is more prominent than that of the electrolytic potential, but the energy consumption of sulfur removal per unit mass does not show a positive correlation. Taking the energy consumption of sulfur removal per unit mass as the efficiency index, better experimental results were obtained under the conditions of 0.8 (V vs. Hg/HgO) anode potential, 50 °C electrolytic temperature, 60 g/L slurry concentration and 14,400 s electrolysis time. The desulphurization rate was 18.85% and the power consumption of desulphurization was 5585.74 W·s/g.

Because of the partial change in the sulfur state, the surface morphology, pore structure and chemical bond of the liquefaction residue are obviously affected by electrolytic oxidation. The lamellar structure has denudation, and the specific surface area and pore volume of liquefied slag particles increase uniformly. Due to the main feature that the deep sulfur in the dense structure of the coal liquefaction residue is difficult to oxidize by contact, deep sulfur has low desulfurization efficiency. Thus, the slurry desulfurization method is more suitable for desulfurization pretreatment under the condition of effective energy consumption utilization. This experimental research method and experimental data can provide a reference for the evaluation of the hydrogen production efficiency of coal liquefaction residue desulphurization coupling.

## Figures and Tables

**Figure 1 molecules-28-02749-f001:**
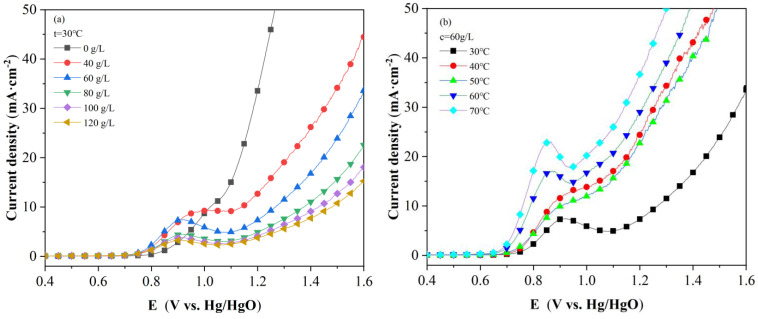
(**a**) LSV curves of SH-DCLR slurry at different concentrations and (**b**) LSV curves of SH-DCLR slurry at different electrolysis temperatures.

**Figure 2 molecules-28-02749-f002:**
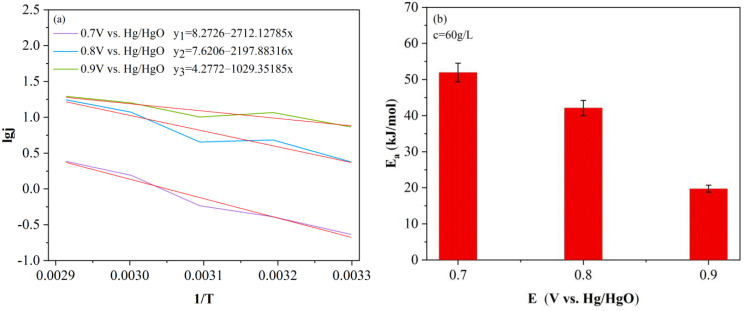
(**a**) Electrolytic dynamics curve of SH-DCLR and (**b**) Apparent activation energy of electrolysis.

**Figure 3 molecules-28-02749-f003:**
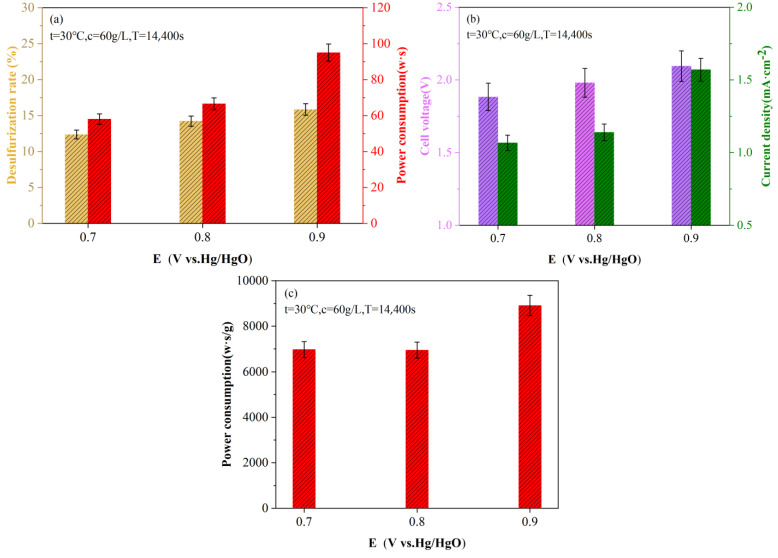
(**a**) Variation diagram of desulfurization rate and total energy consumption under different anode potentials; (**b**) Changes in current density and cell voltage at different anode potentials and (**c**) Energy consumption per unit mass of sulfur removal at different anode potentials.

**Figure 4 molecules-28-02749-f004:**
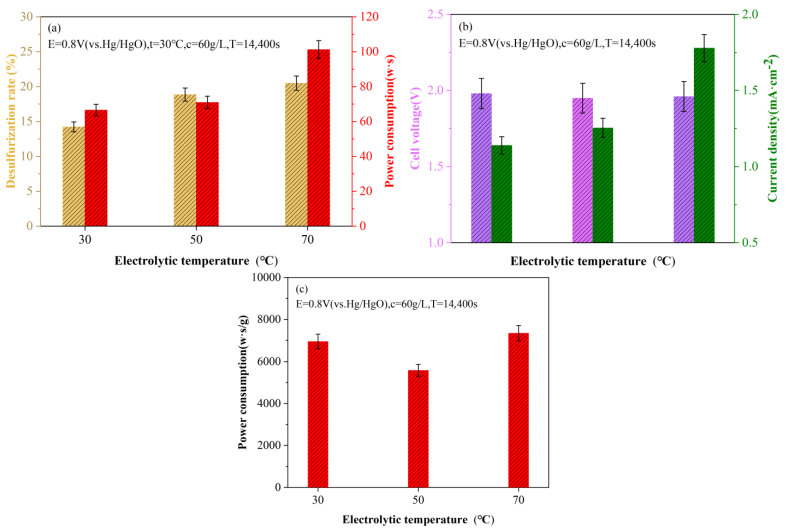
(**a**) Variation diagram of desulfurization rate and total energy consumption under different electrolysis temperatures; (**b**) Changes in current density and cell voltage at different electrolysis temperatures and (**c**) Energy consumption per unit mass of sulfur removal at different electrolysis temperatures.

**Figure 5 molecules-28-02749-f005:**
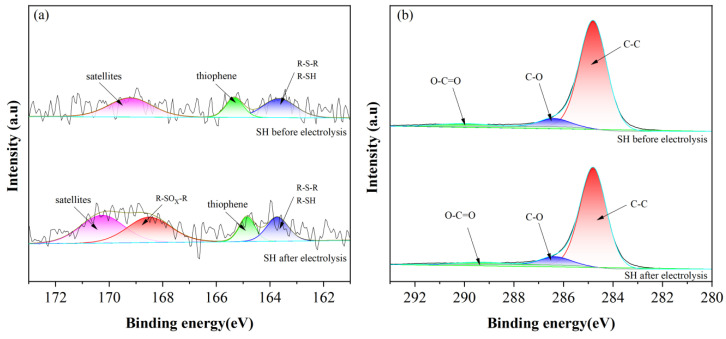
XPS survey of SH-DCLR before and after electrolysis.(**a**) S *2p*; (**b**) C *1s*.

**Figure 6 molecules-28-02749-f006:**
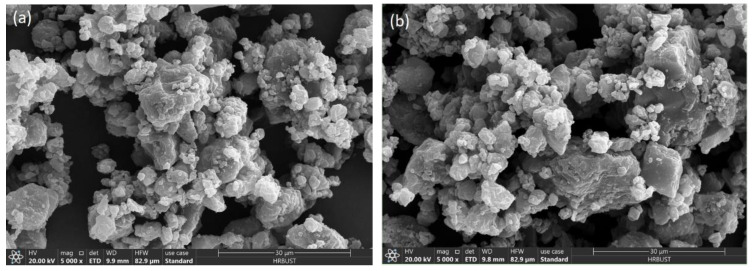
(**a**) SEM images of SH-DCLR before electrolysis and (**b**) after electrolysis.

**Figure 7 molecules-28-02749-f007:**
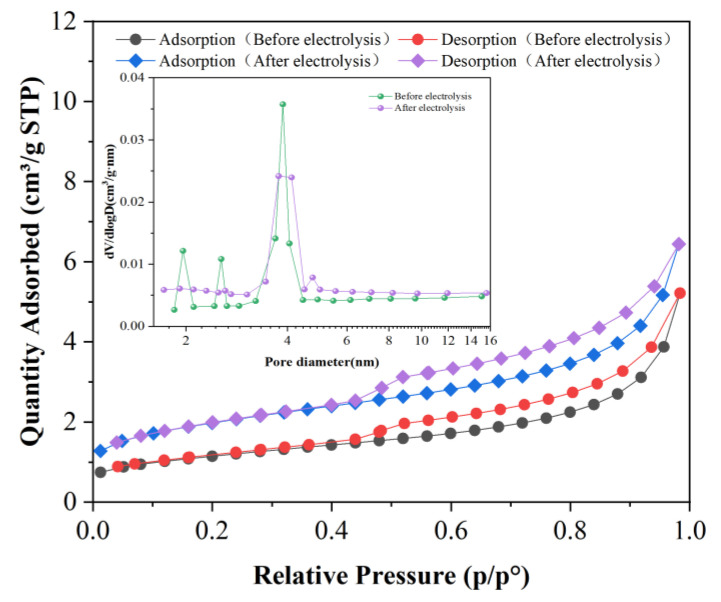
Adsorption–desorption curve and pore distribution of SH-DCLR before and after electrolysis.

**Figure 8 molecules-28-02749-f008:**
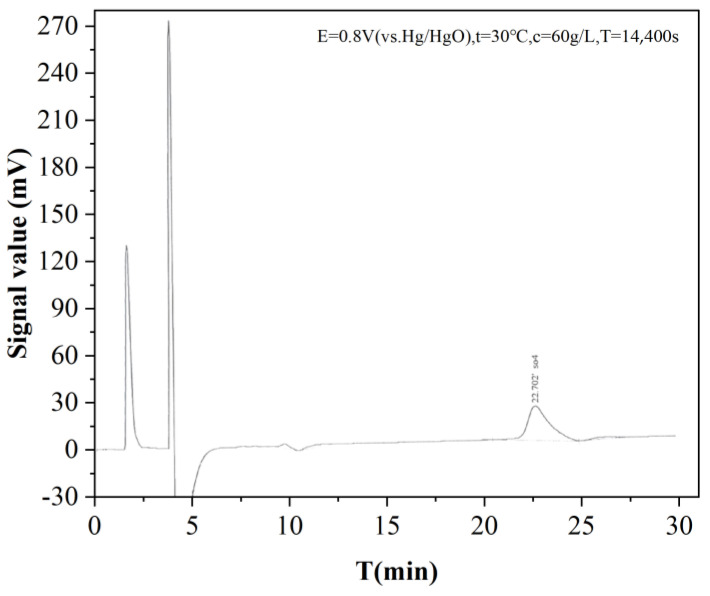
Ion chromatogram of electrolytic solution.

**Table 1 molecules-28-02749-t001:** Sulfur and carbon morphology and relative content before and after electrolysis.

Sample	Relative Content/%						
	Satellites	Thiophene	R-S-R R-SH	R-SO_X_-R	C	C-O	O-C = O
Before electrolysis	46.36	18.90	34.74	0	88.24	7.49	4.27
After electrolysis	35.27	11.94	16.19	36.60	87.46	8.40	4.14

**Table 2 molecules-28-02749-t002:** Porosity characteristics of SH-DCLR.

Sample	Surface Area /(m^2^·g^−1^)	Pore Volume /(cm^3^ g ^−1^)	Average Pore Diameter d/nm
Before electrolysis	3.9825	0.005073	8.1044
After electrolysis	6.7625	0.007413	5.8892

**Table 3 molecules-28-02749-t003:** Proximate and ultimate analyses of SH-DCLR.

Sample	Proximate Analysis Wad/%				Ultimate Analysis Wdaf/%				
	M	A	V	FC	C	H	N	O *	S
SH-DCLR	0.04	15.73	40.53	50.10	76.17	4.38	0.74	16.90	1.81

*: by difference.

**Table 4 molecules-28-02749-t004:** Sulfur distribution analyses of SH-DCLR.

Sample	Sulfur Content/%			
	Total	Organic	Pyrite	Sulfate
SH-DCLR	1.87	1.81	0.06	0.00

## Data Availability

The data presented in this study are available on request from the corresponding author.

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
