# Peer review of "Experimental Study on Electrochemical Desulfurization of Coal Liquefaction Residue"

_molecules, 2023, doi:10.3390/molecules28062749_

Round 1
Reviewer 1 Report
The entitled manuscript “Experimental study on electrochemical desulfurization of coal liquefaction residue” by Zhou and coworkers were reviewed thoroughly. The manuscript is well-written and the authors paid attention and efforts to market their work in this shape. I recommend the manuscript to be published in Molecules; however, some points were found and could be taken into considerations.
In many place in the entire manuscript, a space/blank after the come or full stop (, .) is required.
Page 2-3: in equations 1-8, please use the O letter instead 0 digit for representing oxygen atom. Unfortunately, one cannot follow the equations if they are balance or not.
Line 143 and 145: use the same case of Rho letter.
In all Figures, Descriptions in side the charts are hard to read particularly in Fig 2, 3, 4, 5. Please increase the font size.
Author Response
请参考附件。

Reviewer 2 Report
The presence of sulfur in coal liquefaction residue has an impact on its potential for high-quality and high-value use. Electrochemical desulfurization is characterized by its gentle reaction conditions, ease of operation, simple separation of sulfur conversion products, and minimal impact on the properties of the liquefied residue. To evaluate the efficiency of desulfurization and hydrogen production from coal liquefaction residue, an experiment was conducted using anodic electrolytic oxidation to desulfurize the slurry-state liquefaction residue of a coal-to-liquid byproduct. XPS, SEM, BET, and IC analyses indicated that electrolytic oxidation effectively removed both inorganic and organic sulfur, and that the morphology, pore structure, and chemical bonds of the liquefied residue were altered by this process. This research methodology offers a novel approach and point of reference for evaluating desulfurization efficiency and hydrogen production from coal liquefaction residue. This work can be accepted after a revision.
1. The O's in the chemical equation are all 0, you should correct.
2. Why is the interval of data points set so wide in the author's LSV?
3. Error bars are missing from all of the author's bar charts
4. Since the author has made BET, the adsorption/desorption curve can be presented, while still having the void distribution.
5. At the same time, the author's legend in the figure are too small, it is suggested to increase the font size.
Reviewer 3 Report
Thank you to the authors for submitting their paper in this journal. The paper can make some contributions to the literature but it has some handicaps mentioned below.
1)Introduction section should be rewritten and it should be more comprehensive.
2) The authors should add a paragraph to the introduction section for explaining the contributions of the paper directly.
3) I suggest a literature review section after the introduction section to the authors. In this way they can improve the paper quality.
4) They should check the equations and write the equations according to the journal template.
5) Authors should improve the figure quality and size. In this version I barely read some figures due to the quality and size.
6) They should add a comprehensive discussion section to the paper.
7) In this version there is a very short conclusion section. It should be more detailed and well explained.
Round 2
Reviewer 2 Report
Accept as it is.
Reviewer 3 Report
Authors are solved the mentioned comments.